# Entropy Method for Decision-Making: Uncertainty Cycles in Tourism Demand

**DOI:** 10.3390/e23111370

**Published:** 2021-10-20

**Authors:** Miguel Ángel Ruiz Reina

**Affiliations:** Department of Theory and Economic History, PhD Program in Economic and Business of University of Malaga, Plaza del Ejido, s/n, 29013 Malaga, Spain; ruizreina@uma.es

**Keywords:** information theory, Shannon entropy, forecasting, decision-making, randomness, cycle, tourism

## Abstract

A new methodology is presented for measuring, classifying and predicting the cycles of uncertainty that occur in temporary decision-making in the tourist accommodation market (apartments and hotels). Special attention is paid to the role of entropy and cycles in the process under the Adaptive Markets Hypothesis. The work scheme analyses random cycles from time to time, and in the frequency domain, the linear and nonlinear causality relationships between variables are studied. The period analysed is from January 2005 to December 2018; the following empirical results stand out: (1) On longer scales, the periodicity of the uncertainty of decision-making is between 6 and 12 months, respectively, for all the nationalities described. (2) The elasticity of demand for tourist apartments is approximately 1% due to changes in demand for tourist hotels. (3) The elasticity of the uncertainty factor is highly correlated with the country of origin of tourists visiting Spain. For example, it has been empirically shown that increases of 1% in uncertainty cause increases in the demand for apartments of 2.12% (worldwide), 3.05% (UK), 1.91% (Germany), 1.78% (France), 7.21% (Ireland), 3.61% (The Netherlands) respectively. This modelling has an explanatory capacity of 99% in all the models analysed.

## 1. Introduction

This work aims to estimate the randomness of time series in data-generating processes without prior knowledge for decision-making. The seasonal behaviours of decision-making in time series pose a difficulty in the study of processes. The algorithms proposed in this paper could be used for a wide variety of scientific investigations. However, ignorance of the application of entropy in the social sciences has limited their development and refinement for decision-making applications. In this paper, we provide theoretical and empirical knowledge for the use of entropy as a measurement of uncertainty modelling.

To fully understand the process, it is necessary to develop a theoretical scheme to measure the uncertainty introduced in a nonlinear causal model. Section 2 introduces the work scheme for measuring randomness with a problem to be solved. In particular, agents must decide on two possible choices, thereby giving an idea of the objective pursued by our work, i.e., to determine the uncertainty cycles and the transmission of information between the two stochastic variables to be analysed. To achieve this, the required mathematical and statistical analysis methods are presented in the subsections of Section 2. The metric is based on Shannon Entropy (SE) from Information Theory (IT). The magnitude identifies the temporal randomness in decision-making regarding two possible contemporary events. In Section 2.1, we study linear or nonlinear causality; this allows us to identify the variable to be explained among the variables to be analysed. In Section 2.2, we work with the concept of SE and its temporal organisation. This allows us to identify the cycles of uncertainty. Cycles of uncertainty are theoretically studied in the time domain (correlogram) and the frequency domain (periodogram) in Section 2.3. Once the randomness has been measured, Section 2.4 proposes a nonlinear causality model to measure elasticities and introduce SE as a measure of uncertainty. This measure of uncertainty provides unknown information in the data-generating process of the relationships between the variables to be analysed.

The connection between temporal information content and randomness proposed in this work presupposes an understanding of processes that generate unknown data with mathematical correlations without the previous assumptions regarding the dataset. In Section 3, we empirically analyse the work scheme described above. We illustrate how we have worked step by step with data applied to the tourist accommodation market in Spain. In particular, we are interested in studying how randomness affects decision-making and the cycles found between January 2005 and December 2017 for the main nationalities that visit Spain in this period. A forecasting period, from January 2018 to December 2018, is evaluated. The theoretical implications of the measurement of temporary, seasonal cycles and the frequency domain are analysed in Section 4. Finally, in Section 5, the main theoretical and empirical conclusions of this work are presented.

We conclude this introduction with two subsections: In Section 1.1, which examines the importance of the tourism sector in terms of economic growth, we will describe the importance of uncertainty measurements and how they can help stakeholders’ decision-making; in Section 1.2, a literature review is provided, drawing the reader’s attention to find research gaps in the literature.

### 1.1. Tourism Sector, Gross Domestic Product and Randomness in Decision-Making

Growth in the Tourism Industry is a significant factor for economies around the world. The percentage of gross domestic product associated with the Tourism Industry increases every year [1]. For the Tourism Industry, globalisation means opening borders to facilitate mobility among countries [2]. The root of any analysis of the demand for tourist accommodation lies in understanding the behaviour of potential customers by observing theoretical correlations from an orthodox point of view [3]. The issue of decision-making regarding the demand for accommodation in tourist accommodation, and especially the transmission of unobservable information between time series concerning tourist accommodation, has not been a focus of study in recent years [4]. In this sense, a time series related to tourist accommodation could be decomposed into directly unobservable deterministic or stochastic components (such as trend, seasonality, other cyclical and irregular components), linear or nonlinear [5]. Even though knowledge of their factors is decisive in decision-making [6], tourist accommodation markets reflect the consequences of many consumers making lodging decisions influenced by global seasonal components [7]. The modelling of information flows in the tourist accommodation market is a channel to balance the development of offers with potential visitors in destination tourism markets.

The coronavirus crisis has led to unusual mobility restrictions in recent years [8,9]. The analysis carried out in this paper is within a pre-COVID-19 framework. In the analysis, shocks such as pandemics are considered stochastic and exogenous to the model [10]. The applied scientific approach is based on thermoeconomics and data learning. Thermoeconomics may be introduced as a concept from the heterodox school of economics that applies the laws of mechanical statistics to theoretical economics [11,12,13]. According to the proposed approach, we obtain knowledge about the tourist accommodation market based on the time series study; this approach will be developed in the methodology section. Tourism development provides economic and social advantages to destination areas [14], which means that it is the object of study by countries interested in improving their economic conditions by creating commercial and institutional alliances. These alliances are supported by a body of scientific research on tourism demand, with analytical methods drawn mainly from other branches of science [15].

The discretisation of time series is typical in economics, finance or any other area in which time patterns are recognised. Not infrequently, a discretised decision is solved by answering a dichotomous question; at other times, the answers are multinomial. In the Tourism Industry, dichotomous questions regarding transport (airplane or ship) or types of accommodation (hotel or tourist apartment) may be answered. According to the seasonal behaviour of agents, decisions are made one way or another. Therefore, quantifying the uncertainty of unobserved components in hotel accommodation markets provides information regarding behaviour connected to demand [16].

### 1.2. Literature Review

In this section of the literature review, we will focus on the main papers that have analysed economics agents’ decisions and the market for tourist accommodation. The main reason for this approach is that the literature has focused on tourist apartment accommodation, while no cross-information with tourist hotels has been presented. We wish to note how previous studies have focused on causality rather than on a heterodox approach like the one that is the objective of this study [17,18].

Frequently, researchers identify a mathematical and statistical correlation with causality, which can be considered wrong due to the absence of lags as a prerequisite for causality [19]. An essential aspect of the economic and tourism analysis time series is the study of behaviour and the degree of periodicity. Agents applying the Adaptive Markets Hypothesis make their decisions based on past experiences and expectations. Tourist offers and socioeconomic situations influence decisions; agents can decide based on tourism demand factors such as seasonality or weather [20], social-psychological factors [21], a combination of factors [22] and travel decisions at a microeconomic level [23,24,25,26,27,28,29,30,31,32,33,34]. Referring to the literature on accommodation in tourist apartments, researchers have focused on other factors influencing customer decisions, such as images on web pages [35], externalities created by the market and geolocation offers [36] or quality-price [37].

The emergence of a sharing economy [38] since the 2008 crisis has given rise to real competition for traditional tourist hotels. This change has meant that economic agents have diversified their accommodation demands, now favouring tourist apartments, generating random situations that must be measured to predict tourists’ behaviour [4].

According to the evolution of the digital market, the diversification of decision-making regarding tourist accommodation may be broadly divided between hotels and tourist apartments. Concerning decision-making in the scientific literature of tourism, two approaches have been considered [32,33,39]: studies that have tried to determine the variables that influence decisions and theoretical choice models. This study applies innovative concepts associated with thermoeconomics as used for lodging markets. This paper is a novel contribution to the study of entropy and the methodology developed in transmitting information on the switch in tourist demand from apartments to hotels (and vice versa).

After the literature review, we direct our research toward the processes of generating datasets. We wish to model a stochastic, relevant and dynamic decision-making framework for a global economic market, i.e., decision-making concerning tourist accommodation. The commercial agents that operate in these markets distinguish themselves by offering accommodation services. Those offering services are interested in increasing their productivity, while clients are interested in improving their utility function. Under this approach, the specific behaviour of the agents demanding tourist accommodation can be inferred.

We propose a theoretical and empirical framework for information modelling to complete asymmetric information because firms do not know the intentions of the tourism markets. The basic idea is to model situations of seasonal uncertainty regarding decision-making concerning goods and services. Entropy (SE) will allow us to model situations of uncertainty in order to reduce the asymmetry of information resulting from the market. This work contributes to the traditional areas of statistics/econometrics, notably by presenting a methodological development in decision-making based on noisy data [40].

Although our objective is to model seasonal uncertainty situations and obtain uncertainty cycles while avoiding analogies between other methods, the truth is that similarities arise automatically. In thermodynamics, entropy is commonly associated with a degree of order, disorder or chaos. In our analysis, the system will be orderly, or there will be little uncertainty when decisions made by economic agents are inevitable conclusions. For example, suppose a two-person market in which both parties are demanding a type of service. This type of service can be divided into categories A or B if the two agents consume A, with minimal uncertainty.

On the other hand, when one agent demands A and the other B, the uncertainty is maximal. The analogy does not occur because there is a physical law that describes the behaviour of consumers. It could be given as a description of randomness, which supposes an increase in efficiency. The connection between information content and randomness provides a way to assess the level of randomness in a dataset in a purely mathematical sense, without assuming any underlying model or hypothesis regarding the process of generating the data. Therefore, the framework we have developed focuses on establishing cycles of uncertainty and correctly identifying relevant information.

Regarding the SE applied to temporal uncertainty, we underline that it is a measure of uncertainty and that these patterns (usually cycles) can be quantified. Low degrees of uncertainty means that agents will show predictable behaviour in their decision-making. With a high degree of uncertainty, the opposite will occur [4]. This reveals the importance of measuring the transmission of information systems operated by SE and mutual information [41].

An essential aspect of the present model recognises the behaviour and measurement of random cycles. Entropy, correlogram analysis and Fourier analysis make it possible to visualize stochastics cycles of uncertainty in agent decision-making at the microscopic level. This discretisation and knowledge of uncertainty will lead stakeholders to avoid inefficient decisions [42].

In summary, the study makes three main contributions to the existing literature. First, an effort has been made to develop a novel framework incorporating an unobservable component approach to dynamic decision-making modelling. Second, this study includes a novel dynamic approach to tourist decision-making uncertainty cycles regarding their accommodation choices. The literature identifies these random cycles as seasonal flows; we will observe that a real data analysis allows us to identify seasonal behaviours that may be measured [43]. Finally, introducing a new concept of uncertainty allows us to put forward an econometric model that explains the flows in lodging decision-making. The uncertainty factor in econometric modelling makes it possible to identify seasonal cycles for forecasting and control purposes. In particular, the proposed model will be applied to the Spanish case, using data from official sources. In this way, the model can provide information about the past and forecast future demand for accommodation in tourist apartments. All this embodies a contribution to the literature, which mainly focuses on applying aspects of physics to the tourist market.

## 2. Material and Methods

Initially, randomness generates ignorance due to a lack of standardization of the information. In other cases, information can be deterministic and random. Randomness has been studied in different fields by mathematicians, statisticians, physicists and economists, among others. However, the randomness of a time series presents some problems that classical methods cannot overcome. This is a fundamental assumption that proposes to overcome the traditional barriers of orthodox analysis methods, where conventional tourist causality demand is inferred mainly according to economic (income or prices of products and services), sociological (age, sex or educational level) or seasonal variables (environment or temperature) [3,44,45,46,47,48]. In this work, we start from the fact that the previous information is characterised by total uncertainty. With this assumption, we model freely, without having statistical or economic theoretical impositions that limit the study or condition the results. This approach requires the application of algorithms that explain the behaviour of the data, seeking solutions and obtaining crucial conclusions for organisations in the tourism sector. Thus, the modelling that we carried out allowed us to identify patterns of tourist demand in accommodation. With the knowledge generated through the present methodology, it is possible to describe future market decision-making behaviours. Statistical learning provides a context that can be generalized from learning models, thus avoiding preconceived biases from previous studies [49,50,51,52].

Regarding the measurement of uncertainty, we take as our starting point the concept of entropy, specifically the Shannon Entropy (SE) [53], through the discretisation of temporary resources, which the Markov process can represent. In this way, we can calculate the transmission of information generated from one system to another. In the modelling that is described below, we consider discrete-time intervals. However, the results in continuous processes are suitable for both variables [54]. Originally the SE concept was used in Theoretical Mathematics of Communication to describe messages. Subsequently, it was applied to different fields in Thermodynamics, Statistical Mechanics, and Information Theory (IT) to establish order and reduce or eliminate the uncertainty.

This methodological framework complements the destination choice scheme proposed by Um and Crompton [45] and studies based on weather information for tourists’ decision-making [55]. Specifically, this scheme applies to accommodation selection once a destination has been chosen. According to the authors, the choice of the destination is considered the “output” based on the “inputs” of the destination characteristics, such as location, services, environment or climatic conditions. Thus, we consider that tourists have completed the planning phase based on the destination’s cultural, climatic, or service conditions. Therefore, once the destination has been determined, we focus on modelling the cyclical uncertainty of accommodation between hotels and tourist apartments.

The following subsections describe the theoretical framework in which the causal relationships between hotel and tourist apartment demand are described. A description of entropy can be found in specialised publications or tutorials, where theoretical and practical aspects are reviewed [56]. There is also previous research on IT applications in Economics and Econometrics [57]. The concept of Transfer Entropy (TE) is introduced to capture nonlinear relationships between both series (as a complement to Granger-Causality). SE is presented as a function that measures randomness or uncertainty in agent decision-making. A time series of uncertainty will allow us to determine the cycles of uncertainty (using correlogram and periodogram). Finally, we will establish a theoretical econometric model of causality to predict overnight stays in apartments by introducing the uncertainty measure described theoretically in advance. Finally, the model will be used to forecast and control random situations. This theoretical framework will be empirically verified in the empirical results section. Note that in this work, estimations are made according to points of uncertainty; readers can develop methods of statistical inference and interval estimation according to the processes described in [58].

The objective of establishing an iterative calculation and process of the measurement of randomness for forecasting and control can be considered in the following points based on Figure 1:All theoretical models must be verified through practical application, i.e., assessing the usability principle of the model.All modelling must be understandable and identifiable by subclasses following the criteria of parsimony.All data in the fitted model and its estimated parameters should be used for future iterations of the model parameters.Diagnostic checks. All models must be tested and verified. If inadequacies are found, the models must be re-applied to the identification cycle until an empirically adjustable representation of the data occurs.

The following subsections will detail the modelling carried out step by step. In Section 3.1 of the investigation process, the determination of causality is detailed under linear and nonlinear assumptions. In Section 3.2, the uncertainty algorithm defined by the Shannon Entropy is detailed. Once the algorithm has been described and chronologically ordered, Section 3.3 details the correlograms and periodograms to determine the cycles of uncertainty that occur in decision-making (coincident seasonal patterns). Finally, in Section 3.4, a causality model is defined. Subsequently, entropy and cycles allow forecasting and control tasks to be carried out.

### 2.1. Causality Testing: Linear and Nonlinear Relationships

Depending on the source of information, different tests have varying degrees of popularity in the scientific field. Traditionally, economists and researchers from social branches have used the Granger-Causality test. In contrast, engineers and physicists, among others, have worked with the TE test. In the following sections, we present both tests.

#### 2.1.1. Granger-Causality

Granger-Causality [59] is used for a dual purpose: on the one hand, it can be applied to determine the existence of linear causality between the variables $\left(y_{t},x_{t}\right)$ with dynamic models; on the other, it can determine the direction of unidirectional $\left(y_{t}\to x_{t}\;\mathrm{or}\;x_{t}\to y_{t}\right)$ or bidirectional causality $\left(y_{t}⇄x_{t}\right)$. In this way, we can evaluate the econometric models applied in the subsequent steps of the methodology proposed in Figure 1.

A Granger-Causality test was developed with two variables $\left(y_{t},x_{t}\right)$.
(1)$$\ln(y_{t})=b_{0}\ln(x_{t})+\sum_{j=1}^{n}b_{j}\ln(x_{t-j})+\sum_{j=1}^{m}\alpha_{j}\ln(y_{t-j})+\varepsilon_{t}^{\prime }$$
(2)$$\ln(x_{t})=c_{0}\ln(y_{t})+\sum_{j=1}^{n}c_{j}\ln(y_{t-j})+\sum_{j=1}^{m}d_{j}\ln(x_{t-j})+\varepsilon_{t}^{\prime \prime }$$
where $(\varepsilon_{t}^{\prime },\varepsilon_{t}^{\prime \prime })∼white\;noise$. If $b_{0}\neq c_{0}\neq 0$, the model has instantaneous causality, and lagged causality will be found $\sum_{j=1}^{n}b_{j}\neq \sum_{j=1}^{m}\alpha_{j}\neq \sum_{j=1}^{n}c_{j}\neq \sum_{j=1}^{m}d_{j}\neq 0$. The observed statistic is distributed asymptotically [60].

#### 2.1.2. Transfer Entropy

This section is intended to make an empirical contribution to the transfer of information among variables (system of information). The Granger-Causality test cited above allows us to determine the linear relationship between variables. With Transfer Entropy (TE), it is possible to see the information exchange between two systems or series [61]. TE has shown superior results for the analysis of nonlinearity in IT. It has also been shown that in the Gaussian distribution of random variables, TE and Granger-Causality coincide [62].

To perform an empirical analysis, Shannon TE can be represented as follows [63]:(3)$$T_{X\to Y}=H(Y_{t}|Y_{t-1:t-L})-H(Y_{t}|Y_{t-1:t-L},X_{t-1:t-L})$$
where X and Y are two random variables, L is the lag, and H(Y) is the SE of Y (same for X). The contrasting hypothesis is similar to Granger-Causality but for nonlinear information from one variable to another (X→Y or Y→X). If the *p*-values are less than 5%, we will reject the null hypothesis, resulting in a nonlinear relationship.

### 2.2. Information Theory: Shannon Entropy

Initially, IT has been used to reconstruct messages with a low probability of error [46]. In our work, the message will be the seasonal behaviour of the tourism series. Discrete time is used in many econometric models, with our proposal being suitable for decision-making [64]. To understand the theoretical exposition, we will first define the SE (The choice of the model in base 2 is the usual choice. Subsequently, the informational gain is measured in bits) between two events in discrete-time:(4)$$H(p_{t})=-\sum_{i=1}^{2}p_{it}\;\log_{2}\left(\frac{1}{p_{it}}\right)$$
Now, i=1 is the choice of hotel and i=2 is the choice as a holiday apartment. The subscript t. refers to the corresponding month analysed (the time-frequency is monthly). It is important to understand that the function $H(p_{i}{}_{t})$ is a function of the uncertainty of occurrence of an event in a temporal moment, and that this is an approximation of the real likelihood of choice. The model can be generalised to n. types of accommodations (example for n=3 (apartment, hotel and camping sites)).

To simplify our study, we will reduce our analysis to two types of accommodation. According to official statistics in Spain, hotels and apartments comprise 72% and 16% of tourist accommodation in Spain, respectively. Campsites (8%), rural tourism accommodation (2%) and lodges (2%) make up the rest. The decision will be reduced to two possible choices:(5)$$p_{1t}=\frac{n_{t}}{A_{t}}$$
(6)$$p_{2t}=\frac{m_{t}}{A_{t}}$$
where $n_{t}+m_{t}=A_{t}$ and the axiomatic properties of probability are met (axiom. 1: $0\leq p_{1t}\leq 1$,$0\leq p_{2t}\leq 1$; axiom. 2: $p(E)=p(p_{1t}+p_{2t})=1$; axiom. 3: we assume that $n_{t}$ and $m_{t}$ are two noncompatible events, with $n_{t}\cap m_{t}=\emptyset$ being verified using $p(n_{t}\cup m_{t})=p_{1t}+p_{2t}$). In this study, we will work with n the number of overnight hotel stays in Spain, the number of overnight stays in tourist apartments.

The concept of IT and entropy provide scientific information about the uncertainty of an event. For a given period, it can be demonstrated that the maximally random situation is equiprobable with $p_{1}{}_{t}=p_{2}{}_{t}=1/2$ [4]. There will be a situation with maximum randomness which is unpredictable, but there will also be situations with different degrees of randomness, so we should order the data series according to complexity [46]. Other assumptions that satisfy the definition of entropy are continuity, symmetry, maximum and additivity. The amount of information from SE could be summarised as follows:
$H(p_{i})$ is a monotonic decrease in $p_{i}$.$H(p_{i})\geq 0$ information is a non-negative quantity (for two events $0\leq H(p_{i})\leq 1$).The uncertainty of $H(p_{i})=0$ is zero, in other words, the event always occurs.$H(p_{1},p_{2})=H(p_{1})+H(p_{2})$. This property is crucial and is verified with the maximum entropy demonstrated above $\left(p_{1}=p_{2}=0.5\right)$.

To extend the concept to uncertainty in a random time series, we define its temporal order or chain of entropy. The ordered sequence of functions $H(p_{it})$ is that we will call the time series of the uncertainty function:
(7)$$H(p_{1}),H(p_{2}),\;\ldots \;,H(p_{T})$$

Once the entropy time series has been defined, fluctuation patterns therein will make it more predictable. This likelihood measure of similar patterns of observations will facilitate the analysis of decision-making. We encourage readers who wish to deepen their knowledge of IT to consult other references [65,66]. The following empirical sections will establish the relationships between tourism demand and seasonality according to country of origin. In general, the researchers have ignored the real probability of distribution in decision-making, and the constructed chain will be a measure of randomness for the rest of the investigation.

### 2.3. Correlogram in the Time Domain and Cycles in the Frequency Domain

One unusual use of a correlogram is to analyse the periods (pseudo-cycles) that occur in the movements of an analysed time series. In a stochastic process, the autocorrelation function is defined as follows:(8)$$\rho_{h}=\frac{\gamma_{h}}{\gamma_{0}}\;\forall \;h=0,\;1,\;2\;\ldots \;,\;H$$
where $\gamma_{h}$ is $\mathrm{cov}(H(p_{t}),H(p_{t-h}))$ and $\gamma_{0}$ is $\mathrm{var}(H(p_{t}))$. The graphic representation of the autocorrelation function is the correlogram, i.e., a graphic representation of the relative importance of the past [67]. In the case of sinusoidal waves, it is possible to obtain a measurement of cyclic waves in which it is the objective to describe the amplitude of the entropy changes that occur in decision-making. In modern quantum mechanics, all possible states represented in an entropy time series are separable by the amplitude of the cycles.

According to the idea of harmonic cycles, any stochastic stationary process may be decomposed into cycles of amplitude and a specific period. Fourier, harmonic or fundamental frequencies are defined as follows:(9)$$\lambda_{j}=\frac{2\pi j}{T}\;j=0,\;1,\;2,\;\ldots \;,\;n$$
where $\lambda_{j}$ is the wavelength and T. is the wavenumber related to angular frequency. This trigonometric expression can be expressed through Fourier representation [68].
(10)$$H(p_{t})=\sqrt{\frac{2\pi }{T}}\left\{a_{0}+2\sum_{j=1}^{n}\left\{a_{j}\cos(\lambda_{j}t)+b_{j}\sin(\lambda_{j}t)\right\}\right\}$$
where $a_{j}^{\prime }s$ and $b_{j}^{\prime }s$ are the Fourier coefficients. The periodogram may be expressed as follows:(11)$$I(\lambda_{j})=|W(\lambda_{j}){|}^{2}=\frac{1}{2\pi T}{\left|\sum_{t=1}^{T}x_{t}\exp(-i\lambda_{j}t)\right|}^{2},\;j=0,\;1,\;2,\;\ldots \;,\;n$$
where $W(\lambda_{j})={\left(2\pi T\right)}^{-1/2}\sum_{t=1}^{T}H(p_{i}{}_{t})\exp(-i\lambda_{j}t)=a_{j}-ib_{j}$ is the discrete Fourier Transform of $x_{1},\;x_{2},\;\ldots \;,\;x_{T}$ in at frequency $\lambda_{j}$.

The periodogram shows the importance of each cycle in the sample variance of the series. In this way, we guarantee the asymptotic lack of bias of the estimator of the spectral density function. However, in the present article, it is sufficient to calculate the wavelengths and the associated time to determine the periods of the uncertainty of the SE.

### 2.4. Causality Modelling

This section will describe a form of contemporary causality modelling, as illustrated in Equation (13). In this modelling approach, the SE is included as an explanatory factor. According to the causality test obtained in the empirical section, the explanatory variable $x_{kt}$ will be hotel overnight stays in Spain (by country of origin k), and the dependent variable $y_{kt}$ will be overnight stays in tourist apartments (by country of origin k).
(12)$$y_{kt}=\mu_{1}x_{kt}^{\beta_{k1}}{\left(H(p_{kt})\right)}^{\pi_{k1}}e^{u_{kt}}$$

The model described is nonlinear regarding the parameters to be estimated. The natural logarithms for the dependent variable can be linearized as follows:(13)$$\ln(y_{kt})=\ln(\mu_{1})+\beta_{k1}\ln(x_{kt})+\pi_{k1}\ln(H(p_{kt}))+u_{kt}$$

The parameter $\beta_{k1}$ represents the elasticity of the dependent variable $y_{kt}$ concerning the exogenous variable $x_{kt}$ (see Appendix A). With the introduction of SE $H(p_{kt})$ in the econometric model, the uncertainty in the contemporary modelling of variable $y_{kt}$ is reduced. Parameter $\pi_{k1}$ represents the elasticity of the dependent variable $y_{kt}$ for entropy $H(p_{k}{}_{t})$ and $\mu_{1}$ is the regression intercept. The parameters have been estimated under the ARMA Maximum Likelihood method (OPG—BHHH) for both types of models. The main objective of this type of modelling is to obtain efficient estimators, for which contrasts of non-correlation, called Q-statistic and no ARCH-type effect, will be applied. In this way, we will determine the immediate effect of the uncertainty and whether the predictions have low degrees of variance.

## 3. Results

We obtained information from Spanish official statistics sources and calculated entropy functions. In order to do this, we assumed that the agents presented a function of constant utility and that prices were exogenous and budgetary restrictions existed. Accordingly, we studied how a pure exchange of services occurs. Under this hypothesis, we understand that economic agents make decisions regarding price. However, some studies, besides revealing the importance of budget restrictions, emphasize nonfinancial variables and barriers (sex, age, social and health status) in the tourism market [69].

To empirically test the framework described above, we first present the randomness framework for tourist accommodation in Spain by the country of origin of tourists. The sample analysed was from January 2005 to December 2018 from the Hotel Occupancy Survey collected by the National Statistics Institute of Spain (INE).

Data were extracted for the main nationalities engaging in tourism in Spain: United Kingdom (UK), Germany, France, The Netherlands, Ireland and worldwide (i.e., other countries than those cited here). No missing values were found. The evolution of the series and the observed randomness in accommodation decision-making are analysed in the following subsections.

### 3.1. Causality Testing

Based on the linear causality relationship (Granger-Causality test), hotel accommodation gives rise to accommodation in apartments. According to economic theory, the causality analysed is not traditional in nature, i.e., where cause-effect relationships predominate, nor is there a physical law that theoretically describes this event. However, we consider that this analysis is relevant and that it provides value for decision-making in statistical learning. From a practical point of view, the analysed dynamic correlation means that hotel accommodation flows to generate a secondary market, i.e., accommodation in tourist apartments. This implies that tourists initially stayed in hotels, but they chose to stay in apartments due to the significant demand in the analysed market. As such, apartments now represent real competition to traditional hotel accommodation. This phenomenon led us to choose tourist decision-making as the focus of this paper.

Conversely, it is only empirically demonstrated with 95% confidence that the relationship is bidirectional in the case of the UK. In contrast, regarding the transfer of nonlinear information (TE), it was only possible to demonstrate a potential nonlinear relationship between hotel accommodation and apartments for tourists from the UK (Table 1).

Therefore, from the total number of observations, we can deduce that the information is transferred from the series of hotel accommodation to that of tourist apartments. This establishes a unidirectional relationship in the model.

### 3.2. Randomness Measurement

Having empirically demonstrated a unidirectional causality relationship, we now calculate the temporal randomness. We analysed the randomness of the decision-making of two types of accommodation since January 2005. The nonstandardization of the series implies an initial difficulty in explaining the observed randomness.

In Figure 2, the entropy was selected on a monthly basis and represented in 12 months, starting in January 2012. This aggregation shows systematic behaviour with a seasonal pattern of uncertainty. Regarding nationalities, it should be noted that the most significant periods of uncertainty occurred in months in which the numbers of tourists visiting Spain are generally higher (third quarter). All nationalities showed an increase in entropy in the third quarter, which implies that the decision between renting an apartment or a hotel room was based upon increased seasonal demand. Previously, from February to June, all nationalities showed a decrease in uncertainty levels; as such, demand for hotels was higher than that for apartments. In January, October, November and December, there was a slight rebound in uncertainty. Regarding demand from Irish tourists, entropy was high, i.e., close to 1 throughout the year. As such, it is not easy to determine a priori the accommodation preferences of Irish tourists.

In Figure 2, we can observe common patterns in the series. At different levels, similarities of random oscillations may be observed concerning the global data of Spain. Randomized data showed more significant similarities compared to data from Spain overall regarding tourists from the UK, Germany and The Netherlands. The most remarkable contrasts were observed for visitors from France and Ireland.

### 3.3. Random Cycles

The amount of information and the randomness in the decision-making studied using SE depends on seasonal peaks. In Figure 3, it is possible to observe the cycles produced without distinction among the main nationalities of tourists. The cycle with the highest relative importance corresponds to a period of 6, with posterity produced by a cycle of 12 and a width of 174. The cycle of least relative importance is for Spain overall, i.e., T=4.

Periodograms and correlograms for tourists of different nationalities, i.e., UK, Germany, France, The Netherlands and Ireland, are presented in Appendix B.

Using the periodogram, it was possible to calculate the power of the cycles that occurred in the series. Using the correlogram, the duration of the cycles was determined. In this way, it was possible to determine the intra-annual length of the seasonal periods of repeated patterns. From the evolution of entropy cycles, several countries of origin had approximately 174 months (i.e., Spain, UK, Germany, The Netherlands, Ireland). All the analysed series showed repetitive behaviour with periodicities of 6 and 12 months. For the 12-month cycles, the nationalities that presented the most significant power were the UK and Germany. For the remaining countries, six-month periodic cycles showed more significant influence. The uncertainty cycles with fewer than six months mostly comprised four-month for Ireland.

### 3.4. Causality Model and Forecasting

Once the entropy cycles had been analysed as a proxy variable for seasonality, log-linearised models were estimated (see Table 2) to determine the monthly demand for apartments for tourists’ countries of origin since 2005.

The elasticity between hotel demand and apartments is approximately unitary for all models. Therefore, increases of 1% in tourist hotel demand imply an increase in the demand of tourist apartments of approximately 1% ceteris paribus. On the other hand, entropy in all models yielded the expected sign (positive). In other words, the proportion of information exchange is similar and direct due to the positive sign. Recall from the methodological section that an increase in entropy in our modelling implies a greater increase in the demand for tourist apartments. With a 95% confidence level, this occurred in all models, and this variable may serve as a substitute for the seasonal dummy variables traditionally used in econometric models. Note that the contrasts served to guarantee the efficiency of the estimators and the possibility of performing the hypothesis with the applied parameters.

In general, we observe that the degree of entropy depends on the month analysed in the time series. We verified whether higher entropy translated into more significant uncertainty in the tourist accommodation market. Mathematically and statistically, this has been demonstrated with high levels of confidence.

From a practical point of view, it has been empirically demonstrated with statistical significance that when the demand for hotel accommodation increases/decreases, the demand for tourist accommodation in apartments increases/decreases by the same proportion, which has a direct implication for stakeholders (primary and secondary tourism industries), since this is an indicator of the evolution of the tourism market.

## 4. Theoretical Implications

The results obtained in this work are relevant to the concept of thermoeconomics. They represent an improved means by which to measure uncertainty in the decision-making of economic agents. This measurement implies an increase in productivity, effectiveness and efficiency in the economic industry under study. This statement is based on the methodological analysis carried out in the present research. Informational entropy represents a centre of gravity in the obtained empirical results. Previous studies on entropy applied to other scientific and unexplored branches in the social sciences have been key to obtain these results. This practical work can be considered the beginning of a branch in the measurement of decision-making of economic agents. Future lines of research may aim at multinomial decision-making and stochastic cycle theory. The following conclusions section details the key results obtained from this study.

### Theoretical Implications: Time and Frequency Domain

To demonstrate the validity of the proposed framework, this section presents its performance with simulated datasets generated by a stationary time series model. The simulation tries to mimic some relevant features related to the scheme described above, including decisions and uncertainty.

Economic agents must decide on what is to be consumed; the choice of a particular commodity excludes other possibilities. This decision provides a measure of randomness through entropy and the developed theoretical-methodological analysis. For example, consumers of tourist accommodation (hotels vs. apartments) have to make accommodation decisions; assuming the same preconditions, exogenous factors can explain the final decision.

In this part of the theoretical analysis, an example is presented of a situation of maximum entropy in decision-making, whereby $p_{1}=p_{2}=0.5$ [4]. The duration of observations included in this theoretical dataset is five years (T=60), and a monthly period cycle has been created. The maximum entropy situation would occur in the first month of each year, and the least entropy would happen in the final month of each year.

In the theoretical analysis of the correlogram, we can observe a damped sine wave with a frequency of 12 months. This denotes the repetitive seasonal behaviour with a periodicity of 12, and reveals that situations of uncertainty are cyclical (Figure 4).

As a complementary analysis, we have represented the periodogram to determine the cycles according to frequency. Therefore, we evaluate as relevant those cycles that exceed $\pm 2{\hat{\sigma }}_{I(\lambda j)}$. The existence of a significant period is easy to intuit.

Based on causal relationships, we can establish theoretical relationships regarding the transmission of information from a system x to y (hotels to apartments), which can be interpreted as an increase in uncertainty. Regarding the statistical modelling of causality, equation $y_{kt}=\mu_{1}x_{kt}^{\beta_{k1}}{\left(H(p_{kt})\right)}^{\pi_{k1}}e^{u_{kt}}$ is defined by parameters and variables. These parameters imply the existing relationship between the dependent variable and the explanatory model. The expected signs are as follows: $\beta_{k1}\geq 0$ and $\pi_{k1}\leq 0$. The positive sign means that both variables have a positive theoretical correlation, while the negative sign implies exchanging information in decision-making.

## 5. Conclusions

This research was motivated by the need to understand better the unobservable cyclical uncertainty components in tourist accommodation regarding the choice between apartments and hotels. This is important, given the appearance of a secondary market (tourist apartments) for tourist accommodation in Spain or another international market. At this point, it should be remembered that the present study was carried out in a context in which there was complete mobility of people across national borders. The COVID-19 pandemic has reduced international travel.

The theoretical and empirical results of this research are of potential value to the Tourism Industry. First, we present a new statistical tool for describing decisions regarding agent uncertainty for a set of time-series data. The framework in this investigation consists of defining the decision-making procedure in a dichotomous environment. Decision-making modelling allows researchers to obtain knowledge and quantifications under the Adaptive Markets Hypothesis in the tourist accommodation sector. Second, the empirical results described in the causality testing (Figure 1) show that the causality relationships in the data applied to the tourism sector are unidirectional, i.e., tourism demand for hotel accommodation increases accommodation in tourist apartments in a linear manner, with a 95% confidence level (it is only bidirectional in the case of UK). The direction of causality is a critical element in statistical modelling in determining endogenous and exogenous variables. Knowledge that overnight stays in tourist apartments are more pleasant is a factor that gives empirical added value to the scientific literature.

Third, the uncertainty measure used in this work scheme is the SE time series. This measure from the extraction and classification of messages means overcoming the classic limitations of the probability functions commonly used in Statistics. We found maximum randomness when probabilities were close to 0.5, in agreement with previous studies [4]. This measure of temporal randomness makes it possible to identify the seasonal behaviour of the series that may be systematic according to the nationality under study. According to empirical results, two nationalities stand out, one with high continuous uncertainty and the other with annual increments.

Fourth, once the uncertainty measure has been described, two tools are used to describe uncertainty pseudo-cycles in agents’ decision-making at the microscopic level (discrete-time using a correlogram and frequency domain using a periodogram). This description of pseudo-cycles allows us to determine the natural evolution of the uncertainty series, showing the patterns of intra-annual and annual behaviour in decision-making in situations of uncertainty. The repetitive cycles of uncertainty analysed (Appendix B) describe seasonal movements with a significant periodicity of 6 and 12 months, respectively, for all the nationalities described, with a four-monthly period being noteworthy for Ireland (quarterly). Concerning longer time-frames, all series include a sequence of at least 174 periods.

Fifth, we integrated the uncertainty factor of SE into a causal model, obtaining relevant empirical results on elasticities and uncertainty factors that express cyclical fluctuations, which are typical of decision-making. The models analysed for the main nationalities who visit Spain should highlight the concept of unitary elasticity between demand for hotel accommodation and demand for housing in tourist apartments, as demonstrated in Appendix A. This means that increases of approximately 1% in demand for accommodation in tourist apartments are reflected by an increase in demand for hotel accommodations of roughly 1% ceteris paribus. Regarding the SE factor, the results are caused by the uncertainty cycles described above, highlighting the inverse relationship between the uncertainty factor and the endogenous variables. This may be because decision-making is dichotomous, where increases in one variable imply decreases in the other. For example, empirically, it has been shown that increases of 1% of uncertainty cause an increase in apartment demand of 2.12% (worldwide), 3.05% (UK), 1.91% (Germany), 1.78% (France), 7.21% (Ireland), 3.61% (The Netherlands) respectively. This modelling has an explanatory capacity of 99% in all the analysed models.

Furthermore, the causality models of the time series have a forecasting and control purpose that, in our model, was measured by applying the Theil inequality index. In the example of Spanish tourism, it should be noted that for the training period, i.e., from January to 2018 to December 2018, the results of the Theil inequality index were very close to zero [70].

The scope of the potential for the proposed uncertainty model is vast. In general, the suspicion of uncertainty in decision-making must be measured temporarily; the proposed scheme is very usable in this respect. The dichotomic problem described in this work can be applied to many decisions in the sciences due to continuous decision-making over time (including machine learning or artificial intelligence). For example, analyses could be applied to finance, commodities or transport, in addition to other geographical areas to those discussed in this work. Furthermore, this measure of uncertainty can be compared with binomial-type probability functions. Although causality is the object of the present study, other techniques could be used in artificial neural networks or machine learning to connect causalities for forecasting purposes.

Finally, we should point out that the presented results and suggestions are difficult to interpret without a proposed scheme in Figure 1. Each of the parameters analysed in the scheme can be analysed using techniques deemed appropriate by the scientific community. The knowledge and measurement of uncertainty in decision-making over time can provide a competitive advantage in the new sharing economy, as well as in other fields.

## Figures and Tables

**Figure 1 entropy-23-01370-f001:**
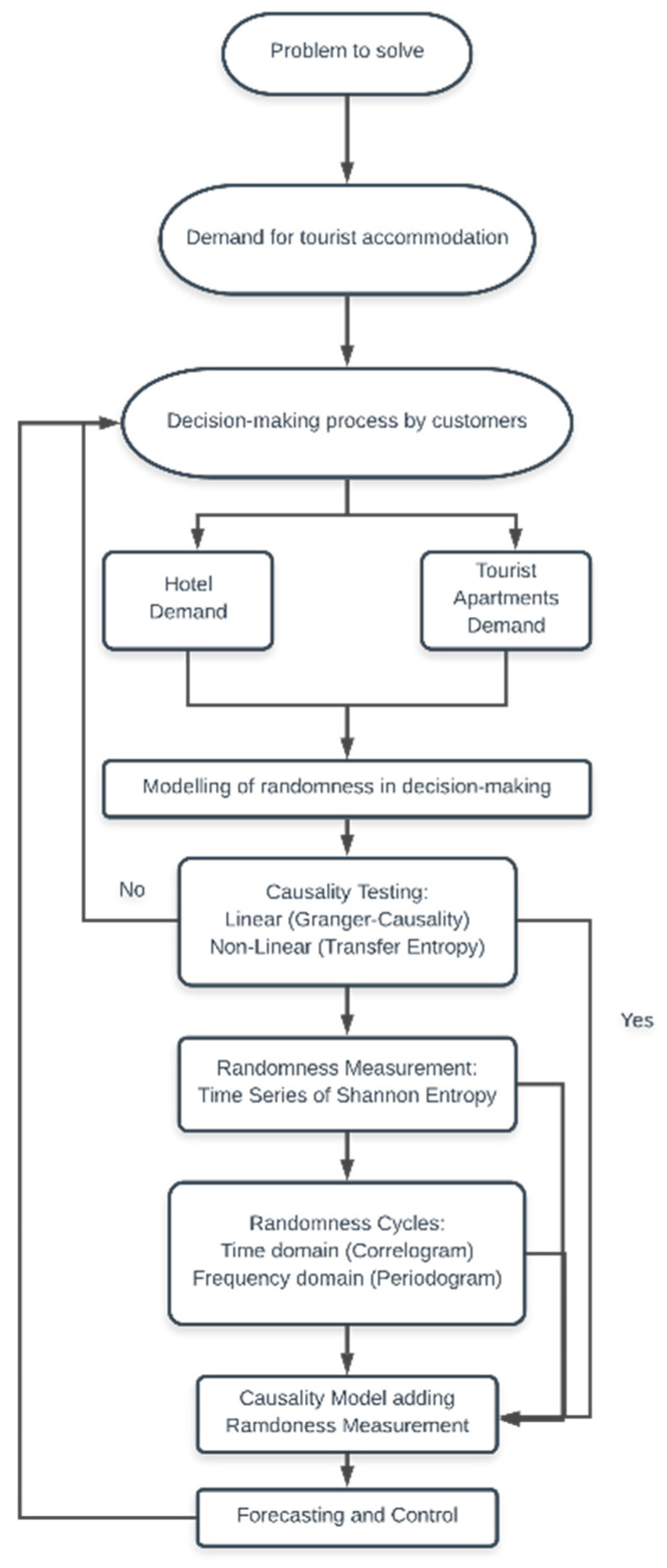
Working scheme in an environment of uncertainty regarding tourist demand: forecasting and control.

**Figure 2 entropy-23-01370-f002:**
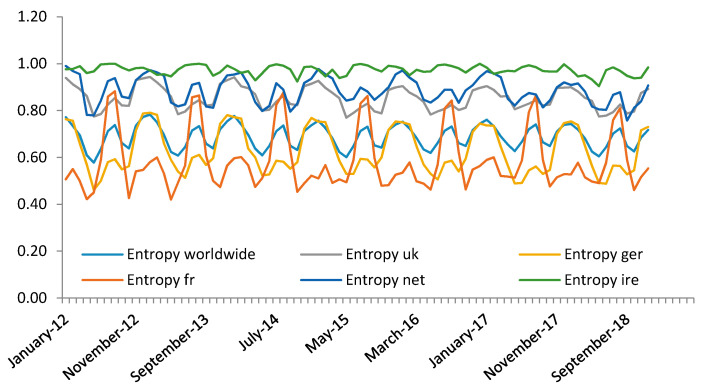
Entropy time series of the main nationalities of tourists visiting Spain. Monthly sample between January 2012 to December 2018.

**Figure 3 entropy-23-01370-f003:**
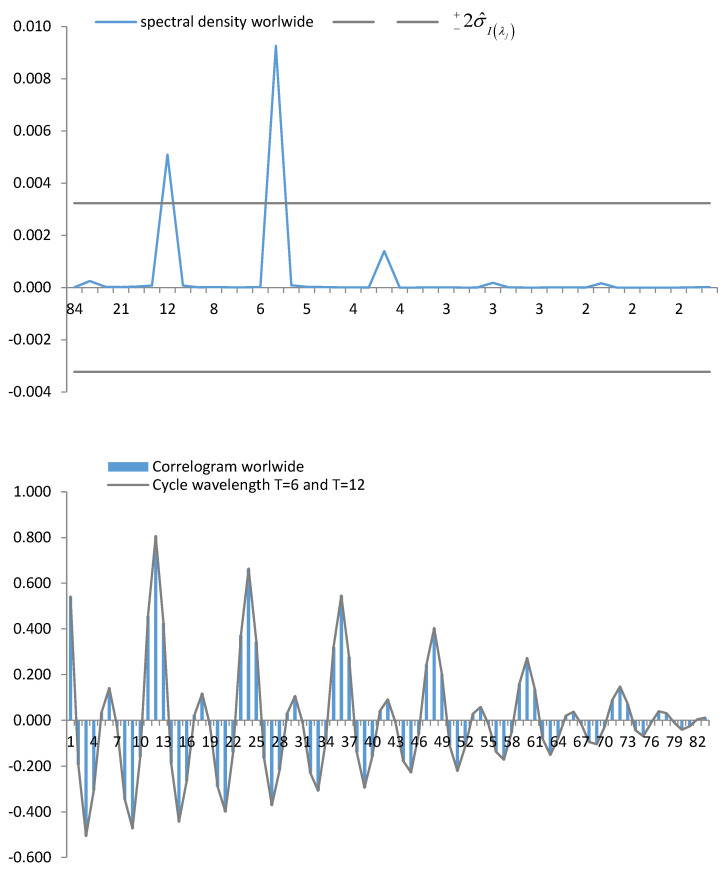
Periodogram (spectral density) and correlogram of entropy for Spain (worldwide).

**Figure 4 entropy-23-01370-f004:**
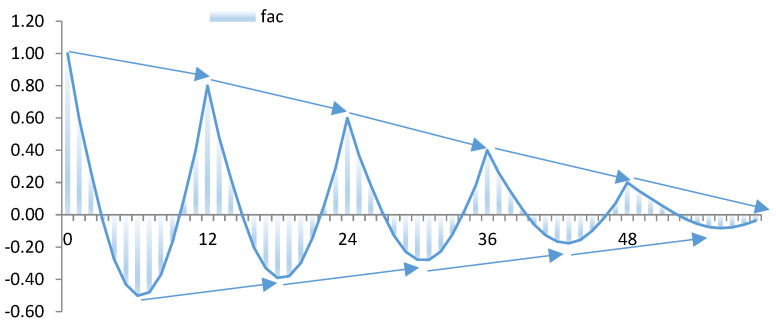
Theoretical decision-making: (1) theoretical cycle of monthly uncertainty. Sample of 5 years and monthly periodicity. (2) Theoretical correlogram (fac) of monthly uncertainty. (3) Theoretical periodogram of monthly uncertainty (spectrum).

**Table 1 entropy-23-01370-t001:** Causality test. Linear and nonlinear relationships: *p*-values.

Direction	Granger-Causality Test	Transfer-Entropy
Spanish Hotels to Spanish apartments	0.000	0.0467
Spanish apartments to Spanish Hotels	3 × 10^−9^	0.5833
UK Hotels to UK apartments	0.0017	0.0000
UK apartments to UK Hotels	8 × 10^−5^	0.1200
German Hotels to German apartments	8 × 10^−5^	0.8533
German apartments to German Hotels	0.1687	0.3567
French Hotels to French apartments	0.0003	0.4400
French apartments to French Hotels	0.1690	0.2000
Netherlands Hotels to Netherlands apartments	0.0065	0.6900
Netherlands apartments to Netherlands Hotels	0.4324	0.1767
Ireland Hotels to Ireland apartments	0.0111	0.0867
Ireland apartments to Ireland Hotels	0.1229	0.5100

**Table 2 entropy-23-01370-t002:** Log-log model for tourist apartment demand by the nationality of tourists. Period of training from January 2005 to December 2017. Forecasting period from January 2018 to December 2018 (*** *p*-value = 0.01). Summarized by the authors, based on data from the National Institute of Statistics of Spain.

	log(Spain)	log(UK)	log(Germany)	log(France)	log(Ireland)	log(The Netherlands)
**Variable**	Coeffi.	Coeffi.	Coeffi.	Coeffi.	Coeffi.	Coeffi.
**C**	0.49 ***	0.06	0.25	0.75 ***	0.40	−0.01
**LOG(hotel)**	1.01 ***	1.03 ***	1.04 ***	1.01 ***	0.98 ***	1.03 ***
**LOG(entropy)**	2.12 ***	3.05 ***	1.91 ***	1.78 ***	7.21 ***	3.61 ***
**AR(12)**	0.92 ***	0.83 ***	0.88 ***	0.99 ***	0.61 ***	0.69 ***
**MA(1)**	0.63 ***	0.45 ***	0.53 ***	-	-	-
**R-Squared**	0.9999	0.9998	0.9998	0.9995	0.9962	0.9976
**Q-Stat (up to 12)**	17.91	6.91	13.96	19.53	6.92	16.18
**ARCH (12) Chi-square**	10.98	9.97	12.30	12.01	0.51	12.85
**Theil ineq. (Jan–Dec 2018)**	0.001082	0.005918	0.002169	0.002169	0.003337	0.017742

## Data Availability

INE: Instituto Nacional de Estadística. Data downloaded by country of origin for hotels (https://www.ine.es/jaxiT3/Tabla.htm?t=2038); and apartments (https://www.ine.es/jaxiT3/Tabla.htm?t=1998&L=0). Accessed on 13 October 2021.

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
