# Peer review of "Entropy Method for Decision-Making: Uncertainty Cycles in Tourism Demand"

_entropy, 2021, doi:10.3390/e23111370_

Round 1
Reviewer 1 Report
Dear author,
I would like to make some comments and suggestions as follows:
- the author should read carefully the INSTRUCTIONS FOR AUTHORS (https://www.mdpi.com/journal/entropy/instructions) and follow them. Thus, the structure of the paper should be modified accordingly.
- in the ABSTRACT section, the author should be more specific regarding the purpose of your study. Please, indicate the main conclusions of your paper. This section should be restructured according to the requirements displayed in the INSTRUCTIONS FOR AUTHORS.
- the words "randomness", "cycle" and "tourism" (instead of "hospitality") seem to be relevant to your study and may be added to the KEYWORDS.
- in the INTRODUCTION section the author should clearly emphasize the importance of his research and the main aim of the paper- it is not obvious what he would like to achieve. The working hypotheses are missing. The LITERATURE REVIEW section should be incorporated in this section. Some paragraphs from this section should be moved to the MATERIALS AND METHODS section (e.g., lines 64-82, 111-114). This section should be restructured according to the requirements displayed in the INSTRUCTIONS FOR AUTHORS.
- in the MATERIALS AND METHODS section the author should be clearly explain why the methodology is appropriate to achieve the aim of the paper and to test the working hypotheses.
- in the DISCUSSION section the author should analyze the results in relation with other previous studies and present the limitations of his research.
- the paper contains some errors (e.g., "the authors"- line 51, line 205). The style is not always appropriate (e.g., line 50, line 174).
All in all, the paper should be improved and restructured.
Good luck!
Author Response
Please see attached pdf document. The document includes: - Comments for reviewers. - Modified paper according to the reviewer's instructions.
Reviewer 2 Report
I like the idea of new applications to decision making in the context of social and economic problems. This is why I rather encourage this line of work. However, I also feel that there much more work to do as for now. Here are my suggestions:
1) The motivation needs to be much better outlined. And more specific, i.e. addressing a gap in the literature.
2) The theoretical framework is rather sketched. I am not suggesting proposing a full structural model, but some clear model should be put forward.
3) I don't find the different statistical tools convincingly combined.
Author Response

(The authors gave the same response as above.)

Round 2
Reviewer 1 Report
Dear Author,
Here are my comments and suggestions:
- as I previously recommended, the author should follow the INSTRUCTIONS FOR AUTHORS. In this respect, the ABSTRACT should emphasize the findings and conclusions and should not contain examples (lines 17-19). In the RESULTS AND DISCUSSION section the author should discuss the results and how they can be interpreted in perspective of previous studies- there is only one reference (line 409)
- some English changes are required (e.g., unfinished phrase- lines 8-9).
Good luck!
Reviewer 2 Report
I found that the authors carried a thorough revision. The suggested changes by the authors have been made.